# Psychological characteristics and the mediating role of the 5C Model in explaining students' COVID-19 vaccination intention

Annelot Wismans[1,2]*, Roy Thurik[1,2,3], Rui Baptista[4], Marcus Dejardin[5,6], Frank Janssen[5], Ingmar Franken[2,7]

**1** Erasmus School of Economics, Erasmus University Rotterdam, Rotterdam, The Netherlands, **2** The Erasmus University Rotterdam Institute for Behavior and Biology (EURIBEB), Rotterdam, The Netherlands, **3** Montpellier Business School, Montpellier, France, **4** CEG-IST, Instituto Superior Técnico, University of Lisbon, Lisboa, Portugal, **5** Université Catholique de Louvain, Louvain-la-Neuve, Belgium, **6** Université de Namur, Namur, Belgium, **7** Erasmus School of Social and Behavioural Sciences, Erasmus University Rotterdam, Rotterdam, The Netherlands

* wismans@ese.eur.nl

**Data Availability Statement:** The survey data that support the findings of this study are available from

## Abstract

To achieve herd immunity against COVID-19, it is crucial to know the drivers of vaccination intention and, thereby, vaccination. As the determinants of vaccination differ across vaccines, target groups and contexts, we investigate COVID-19 vaccination intention using data from university students from three countries, the Netherlands, Belgium and Portugal. We investigate the psychological drivers of vaccination intention using the 5C model as mediator. This model includes five antecedents of vaccination: Confidence, Complacency, Constraints, Calculation and Collective Responsibility. First, we show that the majority of students have a positive propensity toward getting vaccinated against COVID-19, though only 41% of students are completely acceptant. Second, using the 5C model, we show that 'Confidence' (β = 0.33, SE = 03, $p < .001$) and 'Collective Responsibility' (β = 0.35, SE = 04, $p < .001$) are most strongly related to students' COVID-19 vaccination intention. Using mediation analyses, we show that the perceived risk and effectiveness of the vaccine as well as trust in the government and health authorities indirectly relate to vaccination intention through 'Confidence'. The perceived risk of COVID-19 for one's social circle and altruism, the need to belong and psychopathy traits indirectly relate to vaccination intention through 'Collective Responsibility'. Hence, targeting the psychological characteristics associated with 'Confidence' and 'Collective Responsibility' can improve the effectiveness of vaccination campaigns among students.

## Introduction

The development of a vaccine has been recognized as a crucial means to halt the spread of coronavirus disease 2019 (COVID-19). Since effective vaccines against COVID-19 have been developed [1, 2], the greatest challenge is to ensure sufficiently high vaccination rates to

the EUR Data Repository (doi: 10.25397/eur.
14356229).

**Funding:** The author(s) received no specific
funding for this work.

**Competing interests:** The authors have declared
that no competing interests exist.

establish herd immunity. The estimates of the needed vaccination rates to achieve herd immunity range from 67% to 95% [3–5].

In 2019, the World Health Organization declared 'vaccine hesitancy' one of the top ten threats to global health [6]. Vaccine hesitancy is defined as the refusal or reluctance to get vaccinated despite the availability of a vaccine [7]. Vaccine hesitancy has become more problematic in recent decades [8], with the highest levels of skepticism being found in Europe [9]. In a sample of over 7,000 Europeans, 18.9% of respondents reported being unsure about getting vaccinated against COVID-19, while 7.2% indicated that they will certainly not get vaccinated [10]. Even more pessimistic numbers have been shown in a British and Irish sample, with only 65% and 69% of respondents fully willing to get vaccinated, respectively [11].

Governments and public health agencies must be prepared to address COVID-19 vaccine hesitancy [12]. Given its novelty, much is still unknown about the acceptance and motivation behind COVID-19 vaccination. The COVID-19 vaccines differ from previous vaccines in many respects: development speed, innovativeness of the techniques used, uncertainty regarding the magnitude and extent of its effectiveness, and potential side effects. As vaccination willingness is context-, time-, place-, and vaccine-dependent [13], research on COVID-19 vaccination intention and its antecedents is needed, preferably across a variety of target groups and countries.

Previous literature reports potential barriers to vaccine acceptance at different levels [14], ranging from the political and sociocultural levels to the individual level. At the aggregate level, in addition to factors such as the availability and cost of vaccines [7], trust in health officials, the media and governments play an important role in vaccination intention [8]. At the individual level, studies have, among others, shown the relevance of psychological theories of behavior for vaccine acceptance, like the theory of planned behavior [15–17]. Several models have been developed to integrate previous literature on vaccination behavior, such as the 3C [7], 4C [15] and 5C models [18]. Grounded in previous theoretical models, the 5C model aimed at providing a tool useful for both research and practice, reflecting a broad scope of predictors of vaccination intention and behavior [18]. The model includes five psychological antecedents of vaccination, of which the first one, Confidence, relates to trust in the effectiveness and safety of vaccines, in the system that delivers these and in the motivations of policymakers. Secondly, Complacency reflects the perceived risk and perceived level of threat of vaccine-preventable diseases. Thirdly, Constraints reflects the structural psychological and physical barriers, such as those related to geographical accessibility, ability to understand (language and health literacy), and affordability. Fourthly, Calculation relates to individuals' engagement in extensive information searching, which can lead to lower vaccination willingness due to the high availability of anti-vaccination information. Finally, Collective responsibility reflects one's willingness to protect others by getting vaccinated by means of herd immunity [18]. The scale designed to assess these five drivers explained more variance in vaccination behavior compared to previous measures that have focused almost solely on Confidence. Moreover, it was shown that the pattern of the most important Cs within the 5C model varies across vaccines, target groups and countries [18].

Regarding COVID-19 vaccination, previous studies have shown that women, younger adults, unemployed individuals and those with a lower socioeconomic status are less likely to get vaccinated [11, 19, 20]. Moreover, it was recently shown that psychological profiles play a role: vaccine-hesitant and vaccine-resistant individuals are less altruistic, conscientious, more disagreeable, emotionally unstable, and self-interested than are vaccine-acceptant individuals [11]. Finally, higher COVID-19 vaccination intention is associated with more positive general and COVID-19 vaccination beliefs, as well as higher perceived vaccine efficacy and safety [20–22].

The importance of studying psychological variables to understand vaccination intention and inform effective interventions has been advocated [14]. A deeper understanding of the underlying psychology of vaccine-resistant and vaccine-hesitant groups can enhance the potential effectiveness of the public health messages targeting these groups. In this study, we aim to increase the understanding of COVID-19 vaccination by studying the 5C model and its psychological drivers. Since younger people are less likely to suffer from the negative health consequences of COVID-19 infection [23], it is important to know what the main drivers of getting vaccinated are for these individuals. Based on a sample of university students from the Netherlands, Belgium, and Portugal, we pursue the following four objectives.

*First*, we assess the intention to get vaccinated in our international student sample by using a seven-point scale, ranging from completely resistant to completely acceptant.

*Second*, as shown in previous research, the antecedents of vaccine hesitancy differ across vaccines, target groups and countries [18]. We are the first to study which Cs—Confidence, Complacency, Calculation, Constraints, Collective Responsibility (5C's)–are most important for COVID-19 vaccination intention in a sample of university students.

*Third*, as stressed by the authors of the 5C model, knowing the relative importance of the Cs is just a first step, which should be followed by further exploration of the potential levers of these drivers [18]. Using mediation analyses, we investigate which psychological variables, including COVID-19 vaccine-related and COVID-19-related attitudes and personality traits, affect vaccination intention through the 5Cs. This will improve our understanding of vaccination antecedents and, consequently, for which groups reaching desirable levels of these 5Cs and, thereby, vaccination intention may be problematic. The mediation analyses we performed are summarized in Fig 1. Previous studies have shed light on several bivariate relationships between the 5Cs and psychological constructs [18] (presented by the orange arrows in Fig 1). We study whether these constructs indeed affect vaccination intention through the suggested C. Additionally, we study the new indirect relationships represented by the blue arrows in Fig 1. Direct and total relationships are excluded from Fig 1 for clarity reasons.

*Finally*, integrating all results, we formulate advice for governments and public health officials on which Cs would probably best be targeted, while taking their drivers into account when aiming at increasing vaccination intention among students. Knowing for which students' psychological profiles in our sample the Cs are less likely to be present may facilitate the design of targeted public health vaccination campaigns.

We find that Confidence and Collective Responsibility are most important in explaining COVID-19 vaccination among students of our sample. The perceived risk and effectiveness of the vaccine and trust in the government and health authorities indirectly affect vaccination intention through Confidence. The perceived risk of COVID-19 for one's social circle and altruism, the need to belong and psychopathy traits indirectly affect vaccination intention through Collective Responsibility. Thus, vaccination campaigns targeted at students should aim to increase both Confidence and Collective Responsibility, while considering their underlying psychological characteristics.

## Materials and methods

### Data

For the current study, we make use of data from university students. While we acknowledge this group may not be representative of all young adults, especially in terms of education level, we do believe that this will provide a fairer picture of the drivers of vaccination intention among young adults than studies focusing on the general population. As the severity of the consequences of COVID-19 are largely age-dependent, we expect that motives for

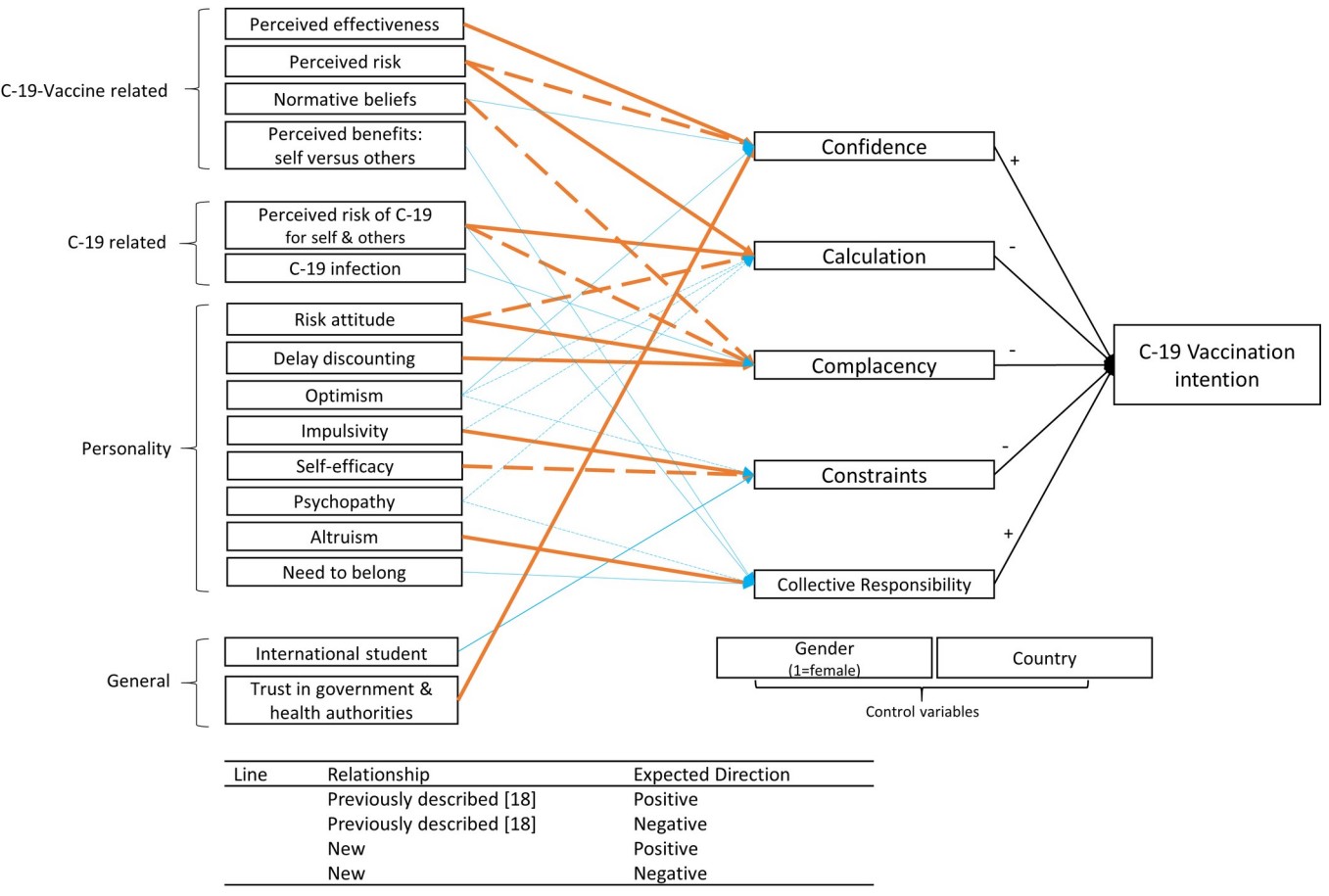

**Fig 1. Overview of expected mediation relationships.** Direct effects are excluded for clarity reasons. (C-19 = COVID-19).

COVID-19 vaccination will strongly differ between older and younger populations. The data used in this study are part of the Erasmus University Rotterdam International COVID-19 Student Survey. This is a longitudinal study on COVID-19-related behaviors and attitudes among university students from multiple countries [24]. Thus far, data have been collected at two points in time. For both data collections, approval was obtained by the Internal Review Board of the Erasmus University Rotterdam. All students signed an informed consent form before starting the survey.

For the current study, we make use of data collected at both moments (T1 and T2) focusing on students from three countries (The Netherlands, Belgium, and Portugal) that participated in both measurement waves. The second survey concentrated on vaccination intention and attitudes.

The first data collection took place during the early days of the pandemic (weeks 17–19, 2020, T1). In total, data from 7,404 university students in ten countries worldwide were collected, amongst which the Netherlands, Belgium and Portugal. At this time, students were approached through university student systems and invitations sent to university e-mail addresses. During this first survey, students could indicate whether they wanted to participate in a follow-up study by sharing their e-mail address. This follow-up study (T2) took place in December 2020 (weeks 51–52). This time, we approached only students from the Netherlands, Belgium and Portugal who participated at T1 and agreed to be contacted for follow-up. Other

country samples were not reapproached since the number of students who agreed to be contacted for follow-up was insufficient to assure large enough samples at T2. Students were contacted through invitations that were sent to the e-mail addresses they provided at T1. In total, 2,902 survey invitations were sent via e-mail at the start of week 51, 2020. Two reminders were sent to those students who did not yet finish or start the survey three and seven days after the first invitation. In total, data were collected from 1,137 students (the Netherlands N = 185; Belgium = 658; Portugal N = 294), for a response rate of 39.2%. This sample is used for the current study. In the analyses, sample sizes can be slightly lower due to the limited presence of missing values and the use of pairwise deletion.

We briefly discuss the data collection method per country at T1 and T2. At T1, Dutch students from the Erasmus University Rotterdam were approached through two university research platforms for students in Psychology and students in Business Administration. For these students it is compulsory to participate in research for a number of hours, and they were thus incentivized to participate in the study. Moreover, the study was shared with all students from the Economics faculty by e-mail. In total, we collected 1,090 responses from Dutch students at T1, of which 633 students (58.1%) shared their e-mail address to be contacted for a follow-up study. 185 Dutch students (response rate = 29.2%) participated at T2. At T1, data from the Belgian sample was collected by systematically contacting all students (around 40,000) via student e-mail addresses from the University of Namur and the Université catholique de Louvain. Students from all faculties and degrees were approached. In total, 3,645 responses were collected at T1, of which 1,660 approved to be contacted for follow-up (45.5%). From these 1,660 students, 658 participated in the second survey (response rate = 39.6%). Finally, the Portuguese students were contacted at T1 by sending invitations to around 9,000 student e-mail addresses of the Instituto Superior Técnico and the Instituto Superior de Economia e Gestão of the University of Lisbon. In total, we collected 1,275 responses at T1 of which 609 agreed to be contacted for follow-up (47.8%), of which 294 participated again at T2 (response rate = 48.3%).

As we did not use a completely probabilistic sample, it should be noted that our findings may not be generalizable to all students. However, we believe that, as we approached representative and large groups of students, risk of bias mostly arises from voluntary participation. It is therefore probable that students who are more agreeable and show more socially desirable behavior are more likely to join in both surveys. To check whether this has affected our outcomes, we conducted all analyses presented in the paper, controlling for scores on the adapted 13-item short (form C) Social Desirability Scale of Marlow-Crowne [25, 26]. The use of social desirability scales has been advocated to check the robustness of results based on self-report data [27]. Based on these additional analyses, we find that all conclusions drawn in the current study remain the same.

At both T1 and T2, surveys were shared using the online survey software Qualtrics. At T1, the survey contained questions on COVID-19-related attitudes, compliance with COVID-19 regulations, and several personality traits. For the current study, only the T1 data on personality traits are used. As personality traits are relatively stable over time [28], we suppose that this is not a problem for the validity of our outcomes. If anything, using multiple measurement times decreases the probability of common method bias [29]. At T2, the survey contained similar questions on COVID-19-related attitudes and compliance with regulations. In addition, questions on COVID-19 vaccination intention and vaccination attitudes were posed. Finally, several personality traits were assessed. The surveys could be completed in English, Dutch or French.

On average, students were 22.92 years old, and 59.3% of the sample was female.

## Measures

The operationalization of all variables is explained in this section.

**Vaccination intention (T2).**   Participants were asked the following question: 'If a coronavirus vaccine that was approved safe and effective was available to you free at cost, would you get vaccinated?' Answers could be given on a seven-point scale: 'definitely not' (1), 'very probably not' (2), 'probably not' (3), 'unsure–neutral' (4), 'probably yes' (5), 'very probably yes' (6) and 'definitely yes' (7). A higher score thus indicates a higher intention to get vaccinated against COVID-19. The continuous scale is used instead of grouping students as being acceptant, hesitant, or resistant. This approach offers a more accurate understanding of vaccination intention, as grouping all students who indicate somewhere between 'probably will not' and 'probably will' under hesitant conditions will lower the unique variation that can be exploited.

**5C scale (T2).**   The 5Cs were assessed using the previously validated 5C scale [18]. The scale consists of 15 items. Each of the Cs—Confidence, Constraints, Calculation, Complacency and Collective responsibility—is captured by three items. Answers are given on a seven-point Likert scale, ranging from 'strongly disagree' to 'strongly agree'. The scale was adapted to specifically focus on COVID-19 vaccinations. A French translation was available [30], while a Dutch translation was performed by two native Dutch speakers individually, after which a consensus meeting took place to discuss and decide on inconsistencies. All items are scored in a way such that a higher score indicates a higher degree of the C assessed. The scores of one of the items of the Collective Responsibility subscale was reversed to be in line with this scoring ('*When everyone is vaccinated, I don't have to get vaccinated too*'). Internal consistency, as reflected by Cronbach's alpha, is acceptable in our sample: Confidence $\alpha = .87$, Complacency $\alpha = 70$, Constraints $\alpha = .69$, Calculation $\alpha = .76$, Collective responsibility $\alpha = .71$.

**Perceived risk of the COVID-19 vaccine.**   Bipolar questions were used to assess the perceived risk of the COVID-19 vaccine. Students were asked the following: 'To what extent do you think the following characteristics apply to COVID-19 vaccines?' Answers could be given on a seven-point scale using bipolar adjectives, which is common practice when assessing attitude [31]. An average score was taken for the following three characteristics: safety ('very unsafe' (1) to 'very safe' (7)), likeliness of side effects ('side effects are very likely' (1) to 'side effects are very unlikely' (7)) and riskiness ('very risky' (1) to 'not risky at all' (7)). The score on safety was reversed before analysis, such that a higher score indicates a higher perceived risk of the vaccine. Internal consistency is very good ($\alpha = .85$).

**Perceived effectiveness of the COVID-19 vaccine.**   A similar question was used to assess the perceived effectiveness of the COVID-19 vaccine. Students were asked the following: 'To what extent do you think the following characteristics apply to COVID-19 vaccines?' Answers could be given on a seven-point scale, ranging from 'very ineffective' (1) to 'very effective' (7).

**Normative beliefs about the COVID-19 vaccine (T2).**   The descriptive social norms in students' social environment regarding getting vaccinated against COVID-19 was assessed using two questions, distinguishing between the norm among family and that among friends. The following questions were used: '*In general, if a coronavirus vaccine that was approved safe and effective was available to your friends for free, what would most of your friends do*?' and '*In general, if a coronavirus vaccine that was approved safe and effective was available to your family for free, what would most of your family do*?'. Answers were given on a scale from 1 (definitely not get vaccinated) to 7 (definitely get vaccinated). An average of the two answers was taken (Spearman's rho = .62, $p < .001$).

**Perceived benefits of the COVID-19 vaccine (T2).**   A question was asked on the perceived personal versus social benefits of COVID-19 vaccination using a bipolar seven-point scale. We asked students to complete a statement—'*Getting vaccinated against the coronavirus*

*will mainly benefit*:', with answer options ranging from 'myself' (1) to '(vulnerable) others around me' (7).

**Perceived risk of COVID-19 for oneself and for others (T2).** Three questions were asked about the risk of COVID-19 for the students themselves. These questions asked about the perceived likelihood of getting infected with COVID-19, getting severely ill if infected and being hospitalized if infected. The same three questions were asked about the risk of COVID-19 for the friends and family of the student. Answers could be given on a seven-point Likert scale ranging from 'no chance at al' (1) to 'absolutely certain' (7). Average values of the three items were taken to create a general COVID-19 risk score for oneself and for others. Internal consistency is acceptable (COVID-19 risk: self $\alpha$ = .67; others $\alpha$ = .71).

**COVID-19 infection (T2).** Students were asked whether they had been infected with the coronavirus before (1 = yes, either confirmed by a test or only expected; 0: no or have not been aware of it).

**General risk attitude (T2).** General risk attitudes were assessed by using the risk propensity scale [32], which consists of seven items. All statements were rated in terms of agreement on a nine-point Likert scale, ranging from 'totally disagree' (1) to 'totally agree' (9), except for the final item, which was rated on a scale ranging from 'risk avoider' (1) to 'risk seeker' (9). Higher scores indicate a higher risk-seeking tendency. Internal consistency was good, at $\alpha$ = .77. A French translation was previously presented based on a back translation approach [33]. The scale was translated to Dutch by two native speakers who first translated the scale individually, after which a consensus meeting took place to discuss and decide on inconsistencies.

**Delay discounting (T1).** Delay discounting is a behavioral measure related to impulsivity and reflects the degree to which people are able to delay rewards, i.e., a measure of impatience. Delay discounting was assessed by the discount rate, with a higher rate reflecting a faster devaluation of delayed rewards and thus greater impulsivity. To capture the discount rate in a fast and accurate manner, the 5-trail Adjusting Delay Discounting Task was used, in which students had to make five consecutive hypothetical choices between receiving €1,000 after a specific delay and receiving €500 directly [34]. The task starts with a delay of 3 weeks, which is increased or decreased based on previous choices. The discount rate is calculated using the hyperbolic discounting model [35] and is log-transformed before analysis, as is commonly done in previous research [34, 36].

**Impulsivity (T1).** The Barratt Impulsiveness Scale-Brief, which is a short unidimensional version of the BIS-11, was used to assess the personality construct of impulsivity [37, 38]. It consists of 8 items scored on a four-point scale, ranging from 'rarely/never' (1) to 'almost always/always' (4). Half of the items were reverse scored. Validated French and Dutch translations were used [39, 40]. The reliability was good, at $\alpha$ = .75.

**Optimism (T1).** Using the Life-Orientation Test-Revised, dispositional optimism was measured [41]. Both Dutch and French translations were already available [42, 43]. The scale consists of 10 items, of which four are filler items. Answers are given on a five-point scale, ranging from 'strongly disagree' (1) to 'strongly agree' (5). Higher scores indicate a higher level of dispositional optimism. Internal consistency was good, as reflected by Cronbach's alpha ($\alpha$ = .81).

**Self-efficacy (T1).** General self-efficacy was measured using the General Self-Efficacy Scale, which was designed to predict individuals' coping with daily hassles and adaptation after stressful events [44]. The scale consists of ten items scored on a four-point scale (1: not at all true; 4: exactly true). French and Dutch translations were available [45, 46]. Internal consistency was very good, at $\alpha$ = .85.

**Psychopathy (T1).** To assess subclinical psychopathy, the psychopathy subscale of the Short-Dark Triad was used [47]. The scale generally consists of 9 items. One item ('*I enjoy*

*having sex with people I hardly know'*) was not included due to cultural controversy. Answers were given on a five-point scale, ranging from 'strongly disagree' (1) to 'strongly agree' (5). Previously made Dutch and French translations were used [48]. Internal consistency was relatively low but acceptable ($\alpha$ = .64).

**Altruism (T1).** The altruism (versus antagonism) subscale of the 100-item version of the HEXACO Personality Inventory-Revised was used, which consists of four questions scored on a five-point scale (1: 'strongly disagree'; 5: 'strongly agree') [49]. Two questions were reverse coded and then transformed; higher scores indicate higher levels of altruism (i.e., being sympathetic and kind). Dutch and French translations were available [50, 51]. Internal consistency was low, at $\alpha$ = .58. Previous studies have found similar low alphas of the altruism subscale while also showing high test-retest reliability and validity [49, 52]. There has been a debate on the relevance of alpha values in evaluating brief personality constructs in such cases [53, 54].

**Need to belong (T2).** The need to belong was assessed using the single-item Need to Belong scale (SIN-B) [55]. It is shown that the SIN-B explains most of the reliable variance of the longer Need to Belong scale [55]. The psychometric properties of the scale are good. Participants indicated to what extent they agreed with the statement '*I have the strong need to belong'* on a five-point scale (1: strongly disagree; 5: strongly agree). A French translation was taken from a French version of the full Need to Belong scale [56], and a Dutch translation was made by two native speakers and decided upon after a consensus meeting.

**Trust in government and health authorities (T2).** Trust in government was measured using the following item: '*In general, how much trust do you personally have in the [name country] government on a scale from 1 (no trust at all) to 10 (full trust)*?' Trust in health authorities was assessed using a similar question and scale: 'In general, how much trust do you personally have in health authorities on a scale from 1 (no trust at all) to 10 (full trust)?' Since the two scores were highly correlated ($r$ = .68), we used an average of the two scores for analyses.

**International student (T1).** We inferred that students who answered 'no' to the question '*Have you lived in [name country] for more than 5 years*?' were international students, which was coded with a value of 1.

**Gender (T1).** Gender was included as a dummy variable, with female (1) and male (0) as answer options.

## Descriptive statistics

The means and standard deviations of all variables and correlations of all variables with vaccination intention and the 5C scale are presented in Table 1 below.

## Methodology

The analyses used are linked to the first three objectives of the study. For the first objective, to assess the willingness to get vaccinated in our sample, the percentage of students who indicated a certain degree of willingness to get vaccinated against COVID-19 were calculated and discussed. For the second objective, studying the link between the 5C model and vaccination intention, one-sided ordinary least squares (OLS) regression analyses were conducted with the 5C subscales as independent variables, vaccination intention as a dependent variable, and country and gender as control variables. We controlled for country differences by including country dummies, and Dutch students were used as a reference group. The standardized coefficients of the regression analysis were used to assess the effect sizes of all Cs to conclude which of these components is most important in explaining COVID-19 vaccination intention among students. Finally, for the third objective, to study the indirect effects of a set of psychological characteristics on vaccination intention through the 5C model, mediation analyses were

**Table 1. Range, Mean (M) and Standard Deviations (SD) of all variables and correlations of all variables with vaccination intention and the 5C scale.**

| Variable (range) | M | SD | 1 | 2 | 3 | 4 | 5 | 6 |
|---|---|---|---|---|---|---|---|---|
| 1. Vaccination intention (1–7) | 5.79 | 1.43 | - | | | | | |
| 2. Confidence (1–7) | 4.97 | 1.48 | .63*** | - | | | | |
| 3. Complacency (1–7) | 2.08 | 1.09 | -.50*** | -.41*** | - | | | |
| 4. Constraints (1–7) | 1.88 | 1.01 | -.47*** | -.49*** | .53*** | - | | |
| 5. Calculation (1–7) | 4.79 | 1.44 | -.29*** | -.32*** | .21*** | .25*** | - | |
| 6. Collective Responsibility (1–7) | 6.04 | 1.08 | .65*** | .56*** | -.59*** | -.51*** | -.24*** | - |
| 7. Perceived Risk C-19 Vaccine (1–7) | 3.57 | 1.32 | -.57*** | -.79*** | .33*** | .45*** | .35*** | -.50*** |
| 8. Perceived Effectiveness C-19 Vaccine (1–7) | 5.17 | 1.20 | .42*** | .66*** | -.34*** | -.35*** | -.20*** | .42*** |
| 9. Descriptive Norm C-19 Vaccine (1–7) | 5.37 | 1.33 | .61*** | .53*** | -.33*** | -.38*** | -.28*** | .45*** |
| 10. Benefits C-19 Vaccine: self vs others (1–7) | 5.45 | 1.41 | -.05 | .04 | .06** | -.02 | .003 | .07** |
| 11. Perceived Risk C-19: Self (1–7) | 3.09 | 0.93 | -0.01 | -.10*** | -.20*** | .03 | .03 | .08*** |
| 12. Perceived Risk C-19: Others (1–7) | 4.23 | 0.92 | .001 | -.06** | -.19*** | -.02 | .04 | .13*** |
| 13. Infection C-19 (0/1) | 0.21 | 0.40 | -.09*** | -.10*** | .12*** | .09*** | .02*** | -.07** |
| 14. Risk attitude (1–9) | 3.69 | 1.24 | -.12*** | -.09*** | .24*** | .07** | -.002 | -.18*** |
| 15. Delay Discounting (ln(.00011)–ln(24)) | -6.11 | 1.78 | -.03 | -.06*** | .08** | .07** | .01 | -.06* |
| 16. Optimism (1–5) | 3.29 | 0.75 | .01 | .12*** | .05 | -.08*** | .03 | .01 |
| 17. Impulsivity (1–4) | 1.96 | 0.46 | -.10*** | -.09*** | .11*** | .06** | -.09*** | -.10*** |
| 18. Self-Efficacy (1–4) | 3.08 | 0.45 | -.01 | .04*** | .05* | -.10*** | .12*** | .03 |
| 19. Psychopathy (1–5) | 1.89 | 0.52 | -.09*** | -.10*** | .21*** | .15*** | .02 | -.16*** |
| 20. Altruism (1–5) | 4.06 | 0.59 | 0.01 | -.03 | -.13*** | -.02*** | .12*** | .13*** |
| 21. Need to Belong (1–5) | 3.40 | 1.03 | .08*** | .01 | -.06* | .003 | .02 | .09*** |
| 22. International Student (0/1) | 0.13 | 0.33 | .02 | .04 | .04 | .06** | .001 | -.03 |
| 23. Trust Government & Health Authorities (1–10) | 6.61 | 1.86 | .43*** | .67*** | -.32*** | -.35*** | -.22*** | .40*** |
| 24. Female (0/1) | 0.59 | 0.49 | -.12*** | -.21*** | -.04 | .05 | .10*** | -.03 |

*: p < .10

**: p < .05

***: p < .01, C-19 = COVID-19

conducted following the procedure suggested by Hayes using the PROCESS macro in SPSS [57]. For each C of the 5C model, three individual regression models were carried out to estimate the indirect effects of the psychological variables expected to be mediated by the C of interest. The first regression model estimated, Model 1, includes the independent variables and control variables, with vaccination intention as the dependent variable. This model presents the total effect of the independent variables (path c, see Fig 2). The second regression model, Model 2, includes all independent variables and control variables, with the mediator as the dependent variable. This model includes path 'a' (Fig 2) and presents the relationship between the psychological variable and the C of interest. Finally, Model 3 is similar to Model 2, but includes—next to the independent variables and controls—the mediator as a predictor, with vaccination intention as the dependent variable. This model contains the direct effect (path c', Fig 2), representing the link between the psychological variable and vaccination intention now controlling for the mediator, and path b (Fig 2), representing the link between the mediator and COVID-19 vaccination intention. Inference on the indirect effect should not be based on the significance of the paths that define it (a and b), but on explicit estimation of the effect by using bias-corrected bootstrapping, which is now considered the standard for testing mediation [58, 59]. Therefore, to estimate the point estimates and confidence intervals of the indirect effects (a*b), we estimated 95% bias-corrected confidence interval (95% BC-CI) using

## Vaccination Intention (%) (N=1,137)

**Fig 2. All paths involved in the mediation analyses, excluding covariates.**

PROCESS. We conclude that indirect effects are statistically significant if the 95% BC-CI excludes zero. As the unstandardized indirect effect cannot be interpreted as a measure of effect size [60], we present standardized indirect effects for all continuous independent variables and partially standardized indirect effects for all binary independent variables [57, 60]. Each of the three regression models were estimated including all the psychological variables expected to be related to a particular C at the same time. Consequently, the direct and indirect effects were estimated whilst controlling for the other predictors of the C. All resulting paths can therefore be interpreted as if they had been estimated simultaneously using simultaneous equation modeling [57]. All data analyses were conducted using IBM SPSS for Windows Version 25.0 [61].

## Results

### COVID-19 vaccination intention among students

Vaccination intention was measured on an ordinal scale, ranging from definitely not to definitely yes. We asked about intention under the condition that the COVID-19 vaccine was approved as being safe and effective and could be received free of cost. Fig 3 shows the percentage per vaccination intention category and cumulative percentages indicated with a dashed orange line (from positive to negative propensity). While the majority of students (85.49%) indicated that they intended to get vaccinated within a range between 'probably' and 'definitely', only 40.9% of the students were totally convinced to get vaccinated ('definitely yes'). Only a very small group was totally resistant to COVID-19 vaccination (1.58%) and indicated that they will 'definitely not' get vaccinated. Almost 1 out of 10 students (9.41%) indicated a negative propensity toward COVID-19 vaccination, as they answered within a range between 'probably not' and 'definitely not'. A total of 5.10% of students indicated being unsure about getting the COVID-19 vaccination and had neither positive nor negative vaccination intention.

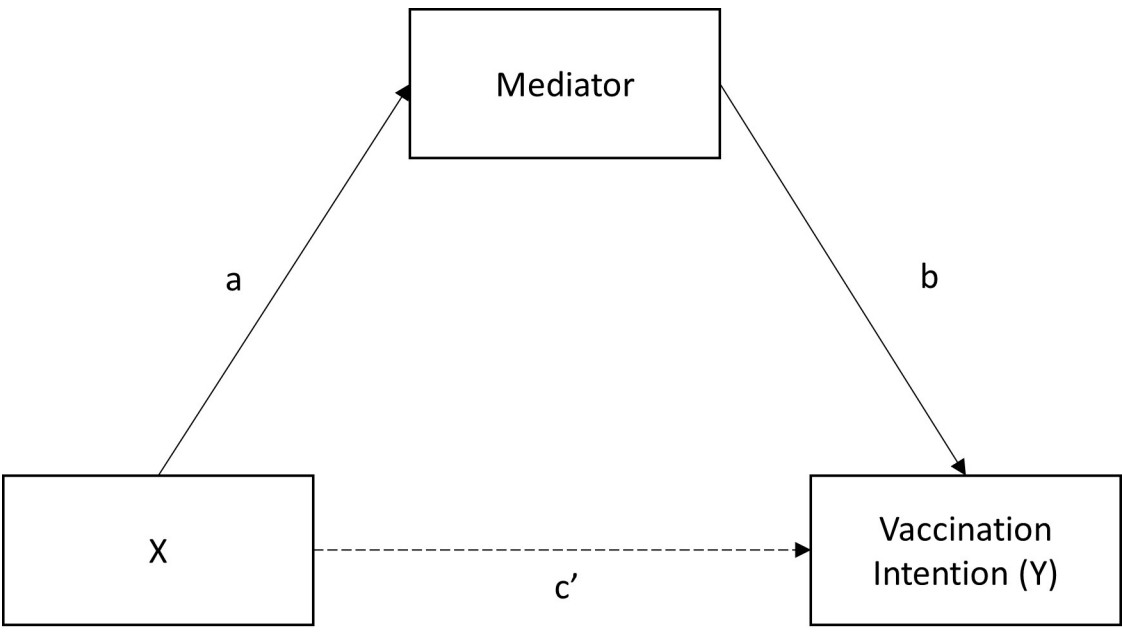

Indirect Effect: c − c' = ab

Direct Effect: c' = c - ab

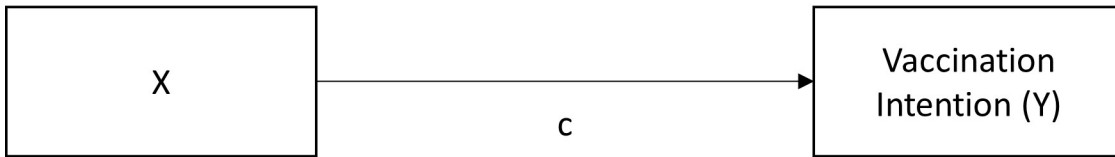

**Fig 3. Vaccination intention in percentages per category and cumulative percentages.**

### 5C model and COVID-19 vaccination intention

Table 2 presents the results of an OLS regression analysis containing the 5Cs as independent variables and vaccination intention as the dependent variable while controlling for gender and country. The regression model shows good fit and high explained variance ($R^2 = 0.54$). Variance inflation factors of the model are all between 1.1 and 2.1, indicating that there is no multicollinearity.

The table shows that all Cs are significantly related to vaccination intention in the expected direction based on the previous literature. Higher Confidence in the vaccine and higher feelings of Collective Responsibility both relate to higher intentions to get vaccinated against COVID-19, while Complacency, Calculation and Constraints are negatively related to COVID-19 vaccination intentions. Relative to the other Cs, the effect sizes of Confidence (B = .32, β = .33, SE = .03, $p < .001$) and Collective Responsibility (B = .46, β = .35, SE = .04, $p < .001$) are largest. We therefore infer that the levels of Confidence and Collective Responsibility play the most important role in explaining the intention to get vaccinated against COVID-19 among students.

**Table 2. OLS regression analysis with vaccination intention (1–7) as the dependent variable.**

|  | B | 95%-CI | β | SE | p |
|---|---|---|---|---|---|
| Intercept | 2.25 | [1.62, 2.88] |  | 0.32 | < .001 |
| **Confidence** | 0.32 | [.27, .37] | 0.33 | 0.03 | < .001 |
| **Complacency** | -0.16 | [-.23, -.09] | -0.12 | 0.04 | < .001 |
| **Constraints** | -0.08 | [-.15, -.003] | -0.05 | 0.04 | .042 |
| **Calculation** | -0.06 | [-.10, -.01] | -0.06 | 0.02 | .009 |
| **Collective Responsibility** | 0.46 | [.39, .53] | 0.35 | 0.04 | < .001 |
| Female (= 1) | -0.11 | [-.23, .01] | -0.04 | 0.06 | .078 |
| Belgium dummy (= 1) | -0.003 | [-.17, .16] | -0.001 | 0.09 | .968 |
| Portugal dummy (= 1) | -0.03 | [-.21, .16] | -0.01 | 0.10 | .788 |
| R² | 0.54 |  |  |  |  |
| F | 163.680 (p < .001) |  |  |  |  |
| N | 1,127 |  |  |  |  |

*Note*: B is the unstandardized beta, and β is the standardized beta. Dutch students serve as the reference group.

## The 5C model as a mediator in explaining vaccination intention

For the third objective, mediation analyses were conducted [57]. Models were estimated for all expected predictors of a particular C at the same time. In this way, we could ascertain the direct and indirect effects of the variables of interest while accounting for the effects of the other predictors of the studied C. In Tables 3–7, the results of mediation analyses are presented, while each table presents the analyses of a particular C.

Fig 4 shows an example of all relationships presented in the tables, using the example of the perceived safety of the vaccine as an independent variable and Confidence as a mediator

**Table 3. Mediation analyses with Confidence as the mediator and vaccination intention as the dependent variable (N = 1124).**

|  | Model 1 | | Model 2 | | Model 3 | | Indirect effect |
|---|---|---|---|---|---|---|---|
| Dependent variable | Vaccination Intention | | Confidence | | Vaccination Intention | |  |
| Paths | c (total effect) | | a | | b and c' (direct effect) | | a*b |
| Coefficient | β | p | β | p | β | p | Indirect effect [95% BC-CI] |
| **Predictors** |  |  |  |  |  |  |  |
| Trust in government & health authorities | 0.11 | < .001 | 0.29 | < .001 | -0.004 | .88 | **0.11 [0.08, 0.14]** |
| Normative beliefs | 0.41 | < .001 | 0.08 | < .001 | 0.38 | < .001 | **0.03 [0.02, 0.05]** |
| Perceived risk of vaccine | -0.29 | < .001 | -0.44 | < .001 | -0.12 | < .001 | **-0.17 [-0.22, -0.13]** |
| Perceived effectiveness of vaccine | 0.07 | .01 | 0.23 | < .001 | -0.02 | .51 | **0.09 [0.07, 0.12]** |
| Optimism | -0.04 | .08 | 0.03 | .08 | -0.05 | .02 | 0.01 [-0.001, 0.02] |
| **Control variables** |  |  |  |  |  |  |  |
| Female (= 1) | 0.03 | .26 | -0.04 | .02 | 0.04 | .07 |  |
| Belgium dummy (= 1) | 0.08 | .01 | -0.05 | .01 | 0.10 | < .001 |  |
| Portugal dummy (= 1) | 0.01 | .63 | -.002 | .28 | 0.02 | .43 |  |
| **Mediator** |  |  |  |  |  |  |  |
| Confidence |  |  |  |  | 0.39 | < .001 |  |
| R² | 0.48 |  | 0.76 |  | 0.51 |  |  |

*Note*: The indirect effects that are bold printed do not contain zero in their 95% bias-corrected confidence intervals (95% BC-CI) and are interpreted as being statistically significant. β is a standardized coefficient. The indirect effect is completely standardized for continuous variables and partially standardized for binary variables.

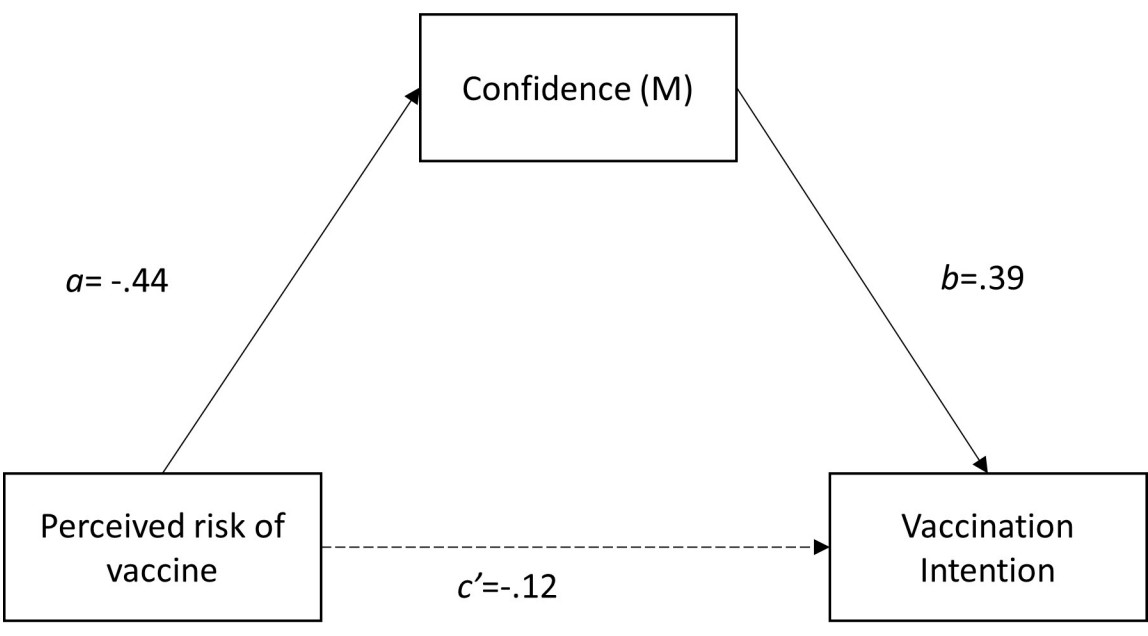

Indirect Effect: c – c' = ab

Direct Effect: c' = c - ab

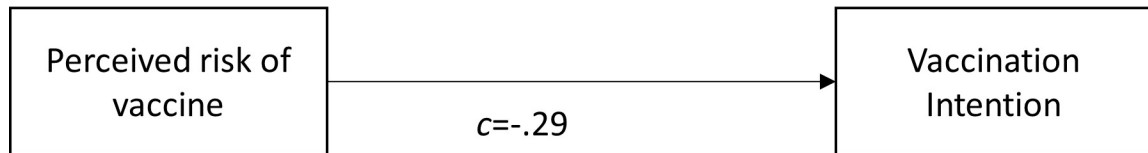

**Fig 4. Example of all paths involved in mediation analyses using the independent variable 'perceived risk of vaccine' and mediator 'Confidence' (Table 3), excluding covariates.**

(Table 3). In Fig 4, we do not show the covariates for clarity reasons, while they are controlled for in the analyses. As shown above, Confidence is strongly positively related to COVID-19 vaccination intention among students. The results of the mediation analyses in Table 3 show that the perceived risk of the COVID-19 vaccine is most strongly associated with vaccination intention through Confidence ($ab$ = -.17; 95% bias-corrected confidence interval (95% BC-CI) = [-.22, -.13]), of which all corresponding relationships are visually presented in Fig 4. Additionally, the perceived effectiveness of the vaccine ($ab$ = .09; 95% BC-CI = [.07, .12]) and trust in the government and health authorities ($ab$ = .11; 95% BC-CI = [.08, .14]) are positively and significantly related to vaccination intention through Confidence. Moreover, a higher descriptive norm (normative beliefs) surrounding COVID-19 vaccination among students' family and friends ($ab$ = .03., 95% BC-CI = [.02, .05]) is also significantly related to higher COVID-19 vaccination intention through Confidence, although the indirect effect is small. Finally, the descriptive norm has a very strong direct relationship with vaccination intention, even after controlling for Confidence ($\beta$ = .38, $p$ < .01).

Table 4 presents the analyses involving Calculation as a mediator. The perceived risk of the COVID-19 vaccine is significantly and negatively related to vaccination intention through

**Table 4. Mediation analyses with Calculation as the mediator and vaccination intention as the dependent variable (N = 1129).**

| | Model 1 | | Model 2 | | Model 3 | | Indirect effect |
|---|---|---|---|---|---|---|---|
| Dependent variable | Vaccination Intention | | Calculation | | Vaccination Intention | | |
| Paths | c (total effect) | | a | | b and c' (direct effect) | | a*b |
| Coefficient | β | p | β | p | β | p | Indirect effect [95% BC-CI] |
| **Predictors** | | | | | | | |
| Perceived risk of C-19: self | 0.06 | .02 | -0.06 | .08 | 0.06 | .04 | 0.01 [-0.001, 0.01] |
| Perceived risk of C-19: others | 0.01 | .76 | 0.03 | .37 | 0.01 | .68 | -0.003 [-0.01, 0.004] |
| Perceived risk of vaccine | -0.57 | < .001 | 0.35 | < .001 | -0.53 | < .001 | **-0.04 [-0.06, -0.02]** |
| Risk attitude | -0.07 | .01 | -0.02 | .53 | -0.07 | .01 | 0.002 [-0.01, 0.01] |
| Optimism | -0.03 | .18 | 0.04 | .20 | -0.03 | .23 | -0.004 [-0.01, 0.002] |
| Impulsivity | -0.06 | .03 | -0.11 | < .001 | -0.07 | .01 | **0.01 [0.01, 0.02]** |
| Psychopathy | 0.002 | .94 | 0.02 | .50 | 0.004 | .87 | -0.002 [-0.01, 0.004] |
| **Control variables** | | | | | | | |
| Female (= 1) | -0.02 | .38 | 0.02 | .47 | -0.02 | .42 | |
| Belgium dummy (= 1) | -0.01 | .77 | 0.03 | .51 | -0.01 | .83 | |
| Portugal dummy (= 1) | 0.001 | .98 | -0.03 | .41 | -0.003 | .94 | |
| **Mediator** | | | | | | | |
| Calculation | | | | | -0.11 | < .001 | |
| R² | 0.34 | | 0.14 | | 0.35 | | |

*Note*: The indirect effects that are bold printed do not contain zero in their 95% bias-corrected confidence intervals (95% BC-CI) and are interpreted as being statistically significant. β is a standardized coefficient. The indirect effect is completely standardized for continuous variables and partially standardized for binary variables.

Calculation (ab = -.04, 95% BC-CI = [-.06, -.02]). A higher perceived risk of the vaccine is related to more Calculation, which is subsequently related to a lower intention to get vaccinated against COVID-19. Moreover, a small indirect effect is present for the level of impulsivity, and more impulsive students show lower levels of Calculation, which is related to lower vaccination intention (ab = .01, 95% BC-CI = [.01, .02]). Other indirect effects, which were expected, are insignificant.

Analyses with Complacency as a mediator are presented in Table 5. All expected indirect effects are significant. Stronger indirect effects are present for the descriptive norm surrounding COVID-19 vaccination among students' social circles (*ab* = .12, 95% BC-CI = [.09, .15]). A higher descriptive norm surrounding COVID-19 vaccination is related to lower Complacency and therefore to higher vaccination intention. Moreover, the perceived risk of COVID-19 for both students themselves (*ab* = .05, 95% BC-CI = [.03, .08]) and for their social environment (*ab* = .05, 95% BC-CI = [.02, .07]) is associated with higher vaccination intention through lower Complacency. Having been infected with COVID-19 is related to higher Complacency and, therefore, lower vaccination intention (partially standardized *ab* = -.05, 95% BC-CI = [-11, -.003]). Students' general risk attitude (*ab* = -.05, 95% BC-CI = [-.08, -.03]) and discount rate (*ab* = -.03, 95% BC-CI = [-.05, -.01]) are also indirectly negatively associated with COVID-19 vaccination intention through higher Complacency.

Table 6 shows the mediation analyses with Constraints as a mediator. We only find a small significant indirect effect of self-efficacy (*ab* = .03, 95% BC-CI = [.003, .07]). Students with a higher level of self-reported self-efficacy perceive fewer constraints, which is related to higher vaccination intention. However, a significant direct effect of self-efficacy on vaccination

**Table 5. Mediation analyses with Complacency as the mediator and vaccination intention as the dependent variable (N = 1128).**

| | Model 1 | | Model 2 | | Model 3 | | Indirect effect |
|---|---|---|---|---|---|---|---|
| Dependent variable | Vaccination Intention | | Complacency | | Vaccination Intention | | |
| Paths | c (total effect) | | a | | b and c' (direct effect) | | a*b |
| Coefficient | β | p | β | p | β | p | Indirect effect [95% BC-CI] |
| **Predictors** | | | | | | | |
| Perceived risk of C-19: self | 0.03 | .33 | -0.15 | < .001 | -0.03 | .33 | **0.05 [0.03, 0.08]** |
| Perceived risk of C-19: others | 0.04 | .13 | -0.12 | < .001 | 0.0003 | .99 | **0.05 [0.02, 0.07)** |
| Normative beliefs | 0.60 | < .001 | -0.33 | < .001 | 0.49 | < .001 | **0.12 [0.09, 0.15]** |
| C-19 Infection | -0.03 | .24 | 0.06 | .02 | -0.01 | .76 | **-0.05 [-0.11, -0.003]** |
| Risk attitude | -0.07 | .003 | 0.15 | < .001 | -0.02 | .40 | **-0.05 [-0.08, -0.03]** |
| Delay discounting | -0.02 | .47 | 0.09 | < .001 | 0.01 | .51 | **-0.03 [-0.05, -0.01]** |
| **Control variables** | | | | | | | |
| Female (= 1) | -0.05 | .08 | -0.04 | .18 | -0.06 | .01 | |
| Belgium dummy (= 1) | 0.02 | .60 | -0.11 | .003 | -0.02 | .49 | |
| Portugal dummy (= 1) | -0.01 | .75 | -0.15 | < .001 | -0.06 | .05 | |
| **Mediator** | | | | | | | |
| Complacency | | | | | -0.35 | < .001 | |
| $R^2$ | 0.38 | | 0.23 | | 0.48 | | |

*Note*: The indirect effects that are bold printed do not contain zero in their 95% bias-corrected confidence intervals (95% BC-CI) and are interpreted as being statistically significant. β is a standardized coefficient. The indirect effect is completely standardized for continuous variables and partially standardized for binary variables.

intention remains after controlling for Constraints (β = -.09, $p < .01$). Optimism, impulsivity and being an international student do not indirectly relate to vaccination intention through Calculation as the confidence intervals corresponding to these variables contain zero.

**Table 6. Mediation analyses with Constraints as the mediator and vaccination intention as the dependent variable (n = 1129).**

| | Model 1 | | Model 2 | | Model 3 | | Indirect effect |
|---|---|---|---|---|---|---|---|
| Dependent variable | Vaccination Intention | | Constraints | | Vaccination Intention | | |
| Paths | c (total effect) | | a | | b and c' (direct effect) | | a*b |
| Coefficient | β | p | β | p | β | p | Indirect effect [95% BC-CI] |
| **Predictors** | | | | | | | |
| Optimism | 0.02 | .62 | -0.05 | .11 | -0.01 | .78 | 0.02 [-0.003, 0.05] |
| Impulsivity | -0.11 | < .001 | 0.03 | .42 | -0.10 | < .001 | -0.01 [-0.04, 0.02] |
| Self-efficacy | -0.06 | .10 | -0.07 | .03 | -0.09 | .003 | **0.03 [0.003, 0.07]** |
| International Student | 0.01 | .64 | 0.06 | .05 | 0.04 | .11 | -0.09 [-0.18, 0.01] |
| **Control variables** | | | | | | | |
| Female (= 1) | -0.10 | < .001 | 0.01 | .63 | -0.10 | < .001 | |
| Belgium dummy (= 1) | -0.08 | .07 | 0.05 | .22 | -0.05 | .16 | |
| Portugal dummy (= 1) | 0.06 | .14 | -0.09 | .03 | 0.02 | .60 | |
| **Mediator** | | | | | | | |
| Constraints | | | | | -0.47 | < .001 | |
| $R^2$ | 0.05 | | 0.03 | | 0.26 | | |

Note: The indirect effects that are bold printed do not contain zero in their 95% bias-corrected confidence intervals (95% BC-CI) and are interpreted as being statistically significant. β is a standardized coefficient. The indirect effect is completely standardized for continuous variables and partially standardized for binary variables.

**Table 7.  Mediation analyses with Collective Responsibility as the mediator and vaccination intention as the dependent variable (n = 1127).**

| | Model 1 | | Model 2 | | Model 3 | | Indirect effect |
|---|---|---|---|---|---|---|---|
| Dependent variable | Vaccination Intention | | Collective Responsibility | | Vaccination Intention | | |
| Paths | c (total effect) | | a | | b and c' (direct effect) | | a*b |
| Coefficient | β | p | β | p | β | p | Indirect effect [95% BC-CI] |
| **Predictors** | | | | | | | |
| Perceived risk of C-19: others | 0.03 | .27 | 0.13 | < .001 | -0.05 | .04 | **0.08 [0.04, 0.13]** |
| Benefits vaccine: self vs others | -0.04 | .13 | 0.05 | .09 | -0.08 | < .001 | 0.03 [-0.01, 0.07] |
| Pyschopathy | -0.10 | < .001 | -0.13 | < .001 | -0.02 | .35 | **-0.08 [-0.13, -0.04]** |
| Altruism | 0.01 | .66 | 0.09 | .01 | -0.04 | .09 | **0.06 [0.01, 0.10]** |
| Need to Belong | 0.14 | < .001 | 0.11 | < .001 | 0.06 | .01 | **0.07 [0.03, 0.11]** |
| **Control variables** | | | | | | | |
| Female (= 1) | -0.14 | < .001 | -0.08 | .01 | -0.08 | < .001 | |
| Belgium dummy (= 1) | -0.14 | < .001 | -0.09 | .04 | -0.09 | .01 | |
| Portugal dummy (= 1) | 0.03 | .41 | 0.06 | .12 | -0.01 | .82 | |
| **Mediator** | | | | | | | |
| Collective Responsibility | | | | | 0.65 | < .001 | |
| $R^2$ | 0.07 | | 0.08 | | 0.45 | | |

*Note*: The indirect effects that are bold printed do not contain zero in their 95% bias-corrected confidence intervals (95% BC-CI) and are interpreted as being statistically significant. β is a standardized coefficient. The indirect effect is completely standardized for continuous variables and partially standardized for binary variables.

Analyses with Collective Responsibility as a mediator are presented in Table 7. We show that the risk of COVID-19 for family and friends, as perceived by students, is positively related to vaccination intention through Collective Responsibility (*ab* = .08, 95% BC-CI = [.04, .13]). Moreover, several personality traits are indirectly associated with vaccination intention through Collective Responsibility. Higher levels of psychopathy traits are negatively related to vaccination intention through lower levels of Collective Responsibility (*ab* = -.08, 95% BC-CI = -.13, -.04]). Conversely, higher levels of altruism (*ab* = .06, 95% BC-CI = [.01, .10]) and the need to belong (*ab* = .07, 95% BC-CI = [.03, .11]) positively indirectly relate to vaccination intention through Collective Responsibility.

## Discussion

According to the results, the majority of the 1,137 Dutch, Belgian and Portuguese students in our sample do not have a full and definite intention to get vaccinated against COVID-19. More than half of them (57.7%) fall on a continuum between leaning toward acceptance and leaning toward resistance. Although a large majority of our sample has a positive propensity toward getting vaccinated against COVID-19 (85% of students indicate intentions between 'probably' and 'definitely'), the group of students who are completely acceptant of the vaccine (41%) is quite small. At the same time, only a very small group indicates to refuse a vaccination (1.6%). To achieve herd immunity through vaccination, it is crucial that more students shift their intention toward a more positive definite answer. Most gains can be achieved by targeting students who already have a positive propensity toward vaccination but are not completely certain. As previous studies mostly use yes/no scales to assess vaccination intention, it is not possible to directly compare our results to those of previous studies. For example, using a yes/no format, 95% of respondents indicate a willingness to be vaccinated against COVID-19 in a sample of students in Italy [62].

### 5C drivers of students' COVID-19 vaccination intention

We show that all five components of the 5C model—Confidence, Calculation, Complacency, Constraints and Collective Responsibility—are related to COVID-19 vaccination among students in our sample. Confidence, i.e., the degree of trust in the vaccine and the system that delivers it, and Collective Responsibility, i.e., the willingness to protect others by getting vaccinated, are most strongly related to COVID-19 vaccination intention. This suggests that campaigns targeted at increasing vaccination intention among students will likely be most successful when focused on enhancing the levels of both Confidence and Collective Responsibility. Smaller negative links are present between vaccination intention and Complacency, Constraints, and Calculation.

### Psychological profiles underlying COVID-19 vaccination intention

We show that psychological profiles indeed play an important role in explaining vaccination intention. As vaccination campaigns will likely be most successful when targeted at Confidence and Collective Responsibility, we discuss which psychological variables underlie these drivers and should therefore be considered when designing interventions.

First, we show that the perceived risk and effectiveness of the vaccine both affect vaccination intention through changes in Confidence levels. We find that the level of Confidence is lower for students in our sample who perceive the vaccine as being riskier (e.g., less safe and with a higher risk of side effects) and less effective. Moreover, trust in the government and health authorities plays an important role in explaining vaccination intention through Confidence. Students with lower trust in these institutions report lower levels of Confidence, which translates into lower vaccination intention. Finally, the descriptive norm in students' environment—the degree to which family and friends intend to get vaccinated—has a small effect on intention through Confidence. Moreover, we show that the descriptive norm also has a strong direct relationship with vaccination intention.

With respect to Collective Responsibility, it is evident that the perceived risk of COVID-19 for people in a student's social circle indirectly relates to his/her vaccination intention through Collective Responsibility. Students in our sample who perceive the risk of COVID-19 for their environment to be low indicate a lower intention to get vaccinated against COVID-19, motivated by a lower willingness to protect others. Moreover, we show that personality plays an important role in explaining the perception of vaccination as a Collective Responsibility. Psychopathy traits, which are related to antisocial behavior caused by deficits in empathy, emotion, and self-control [47], negatively relate to Collective Responsibility and, therefore, to a lower intention to get vaccinated. Similarly, students with more altruistic personalities, e.g., those who feel more sympathy toward others and want to help those in need, have a higher intention to get vaccinated against COVID-19, through higher levels of Collective Responsibility. Additionally, the degree to which students feel the 'need to belong' indirectly relates to higher vaccination intention through Collective Responsibility. The need to belong relates both to the human needs of wanting to affiliate with others and wanting to be accepted by others [63]. We expect that both a need to be in contact with others at risk for COVID-19 without worrying and signaling prosocial behavior to be accepted by others underlie the indirect positive relationship between the need to belong and vaccination intention through Collective Responsibility.

### Implications for vaccination campaigns and interventions

What implications can these results have for public health policy? First, the data suggest that seeking to increase both Confidence and Collective Responsibility simultaneously will be

worthwhile since vaccination interventions that address multiple underlying drivers have been shown to be more successful [64]. We provide several suggestions for both drivers separately.

Based on the findings of our study, in targeting Confidence it is important to influence the perceived safety and effectiveness of the COVID-19 vaccine. In our survey, the most prevalent reasons for not getting vaccinated were related to worries about safety, side effects, development speed and the wish for the vaccine to be proven effective and safe over a longer period. By challenging the misinformation surrounding the vaccine and providing factual information on, for example, the reasons that the vaccine was able to be developed so fast, Confidence in the vaccine can be increased. However, it is important to think about how and who communicates this information because, for people with a strong prior opinion, a correction of information could backfire and lead to even more divided attitudes [65]. Since we showed that low Confidence is related to lower trust in the government and health authorities in our sample, information about safety and efficacy should preferably be communicated by people not within traditional positions of authority. A good strategy would be to use 'surprising validators', i.e., people seen as credible to the target audience but who are not expected to share this information [65]. To reach students, one could, for example, think of campaigns including peers or celebrities.

We find Collective Responsibility to be the strongest predictor of COVID-19 vaccination among students of our sample. It is logical that this is an important driver for this group since students are less at risk of developing severe health consequences if infected by COVID-19. Willingness to protect others by getting vaccinated is thus a strong motivator. We show that the perceived risk of COVID-19 for others in a student's social circle indirectly affects his or her vaccination intention through Collective Responsibility. Students with at-risk family members may thus be more likely to get vaccinated to protect those around them. Vaccination campaigns aimed at students may therefore be more successful by showing the risks for those in the close environment of students. Explaining the concept of herd immunity through vaccination is an important approach, as was also experimentally shown [66]. Students can and should be made aware that they are not just making an individual decision but also a collective decision when deciding whether to get vaccinated. To increase identification, campaigns could discuss reasons why certain groups are unable to get vaccinated (e.g., people with allergic reaction to vaccines, autoimmune diseases or other conditions). Nevertheless, our results also indicate that students in our sample with less altruistic, emphatic, and social personalities were less likely to feel Collective Responsibility. Influencing these personality traits is likely to be very difficult, maybe even impossible. But one should consider that, as these students feel less empathy toward others, campaigns focused on stressing the prosocial consequences of vaccination may not be sufficient to influence certain groups as strongly and could even promote the idea of free riding [67]. Therefore, it remains important to communicate the personal risks of COVID-19 for young adults, for example, by communicating the possibilities of long-lasting adverse consequences of COVID-19, also known as 'long COVID' [68].

In addition to positively affecting vaccination intention through Confidence and Complacency, we show that the descriptive norm has a strong direct relationship with vaccination intention. Descriptive norms have been proven to be strong drivers of behavior, especially in times of uncertainty [69]. Vaccination campaigns may be more successful if they make the norm among students more salient by stressing that the majority of students intend to get vaccinated.

In most countries, young adults will be the last in line for vaccination. Although this makes sense from a health perspective, governments should realize that by the time students must actively decide whether to get vaccinated, the vaccination strategy may have already led to decreased infection rates and, therefore, also to a lower perceived risk of COVID-19. Importantly, when family members are already vaccinated, the level of Collective Responsibility may

decrease through a lower perceived risk of COVID-19 for others. It is therefore vital that campaigns focused on students start early on since the necessity of vaccination is most salient at that stage, and, therefore, positive intentions can be formulated. Studies show that once a strong enough intention to get vaccinated is formed, this likely translates into action [70]. In terms of policy, to enhance the transition from intention to behavior, the process of getting vaccinated should be easy, fast and without unforeseen barriers [71].

## Limitations and future research

The study has several limitations. *First*, we measure vaccination intention and not actual vaccination behavior. As the intention-behavior gap shows us that not all intentions translate into behavior [72], it would be interesting to research whether our results also hold with actual vaccination behavior as the dependent variable. *Second*, as we did not use a probabilistic sample, the use of inferential techniques is not entirely justifiable [73, 74]. While we used a large sample of students from three countries and, during the sampling process, approached large and representative groups of students, participation was (mostly) on a voluntary basis. Since we expected students with higher levels of social desirability to be more likely to participate, we conducted all analyses controlling for social desirability. The fact that our conclusions remained the same strengthen our belief in the validity of our results. However, it is possible that our sample suffers from other type of non-response bias and that our results should therefore be interpreted with caution. *Third*, as discussed, vaccination intention is context- and time-dependent. Since we use a snapshot of vaccination intention assessed in December 2020, attitudes and intention toward vaccination may have shifted over time. *Finally*, for future research, an important next step will be to design and test which interventions have the best outcomes in both experimental and real-life settings.

Despite its limitations, our study provides governments and public health officials with much needed levers of the important drivers of COVID-19 vaccination intention among students. Given the suggested rate of COVID-19 vaccination acceptance in our sample, we hope that our findings will contribute to the designing and improving of effective public health messaging to increase the acceptance above the percentages needed to achieve herd immunity.

## Acknowledgments

We thank Karl Wennberg, Srebrenka Letina, Enrico Santarelli, Andrew Burke, Jinia Mukerjee, José María Millán, Jorge Barrientos Marín, Joern Block and Olivier Torrès for their involvement in the T1 data collection in countries not used in the present study.

## Author Contributions

**Conceptualization:** Annelot Wismans, Roy Thurik, Rui Baptista, Marcus Dejardin, Frank Janssen, Ingmar Franken.

**Formal analysis:** Annelot Wismans.

**Investigation:** Annelot Wismans, Roy Thurik, Rui Baptista, Marcus Dejardin, Frank Janssen, Ingmar Franken.

**Project administration:** Annelot Wismans.

**Writing – original draft:** Annelot Wismans, Roy Thurik, Ingmar Franken.

**Writing – review & editing:** Annelot Wismans, Roy Thurik, Rui Baptista, Marcus Dejardin, Frank Janssen, Ingmar Franken.

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
