## [Decision Letter · Decision Letter 0]

21 Jun 2021

PONE-D-21-15022

Psychological characteristics associated with students’ COVID-19 vaccination intention

PLOS ONE

Dear Dr. Wismans,

Thank you for submitting your manuscript to PLOS ONE. After careful consideration, we feel that it has merit but does not fully meet PLOS ONE’s publication criteria as it currently stands. Therefore, we invite you to submit a revised version of the manuscript that addresses the points raised during the review process.

In the revised version of the paper please focus more on improving and better explaining the statistical analysis as mentioned in the reviewers' comments listed at the bottom of this email.

We look forward to receiving your revised manuscript.

Kind regards,

Camelia Delcea

Academic Editor

PLOS ONE

Journal Requirements:

Reviewers' comments:

Reviewer's Responses to Questions

**Comments to the Author**

1. Is the manuscript technically sound, and do the data support the conclusions?

Reviewer #1: Yes

Reviewer #2: Partly

Reviewer #3: Yes

Reviewer #4: Partly

2. Has the statistical analysis been performed appropriately and rigorously? 

Reviewer #1: Yes

Reviewer #2: No

Reviewer #3: Yes

Reviewer #4: Yes

3. Have the authors made all data underlying the findings in their manuscript fully available?

Reviewer #1: Yes

Reviewer #2: Yes

Reviewer #3: Yes

Reviewer #4: Yes

4. Is the manuscript presented in an intelligible fashion and written in standard English?

Reviewer #1: Yes

Reviewer #2: Yes

Reviewer #3: Yes

Reviewer #4: Yes

5. Review Comments to the Author

Reviewer #1: -the authors should be write the abbreviation lists example: COVID-19,UK,IBM,SPSS etc.

- the authors give a gap or the space appears the line of the text in the references.

-clarify the statistical model used for this study.

- how to determine the sample size and which sampling technique is applied?

Reviewer #2: The authors describe partial results from the Erasmus University Rotterdam International COVID-19 Student Survey (EURICSS), related specifically to vaccination intention in university students from three countries: the Netherlands, Belgium and Portugal. They use the questions related to scales measuring the 5C model and some personality trait questions, along with a measure of vaccination intention, to perform a regression analysis and a mediation analysis.

The paper is written in a clear manner and presents enough details about the definition of each of the variables, and how they were measured in the survey. Two things are of concern in my opinion: 1) the authors are not using a probabilistic sample, or at least it was not explained as such in the methods section, and thus using or calculating standard errors and p-values for the regression model and the mediation analysis is not entirely justifiable. 2) All the figures in the paper need to have a better quality: figures 1 and 3 do not have an appropriate resolution to be readable. Fig 3 does not include a clear label on the x axis. Figs 2 and 4 do not include the paths, and thus are not very useful.

The description of the results per question should be included in the paper. This table is now in the supporting information, but it could be added to the main article without the correlations, and a heat plot can be used to represent the correlations between variables. This description is specially useful when presenting the regression model results. In this case, Table 1 is giving inferential statistics details (that should be calculated only if this was a probabilistic sample), but it is not very illustrative about 1) the assumptions of the model and 2) the complete description of each of the variables.

The mediation analysis are hard to follow without Fig 2 and 4 and/or a model equation. In any case they have the same problem as the regression model: a clear description of the variables (are there extreme values that could be affecting the analysis, for example?) and the fact that the authors are using inferential techniques on what looks like a convenience sample.

In summary, without a probabilistic sample the inferential results should be taken with a grain of salt. The authors do note in the limitations of the study that the sample might not be completely generalizable to all young adults, but they do interpret their results in a matter that leads to believe that they are generalizable to all the students from these three countries. My question is: if this is the case, and if so, how would the authors argue that this is true.

If this is clarified, and the recommendations about the results and figures are implemented, I think this manuscript can be technically sound.

Reviewer #3: Comments

The paper examines an interesting area, as it examines the psychological characteristics associated with students’ COVID-19 vaccination intention. Having read through the paper, I do not believe it needs any grammar editing.

Having read through the article, I have the following comments and suggestions.

Overall

The article has too many words in its current state. The authors should try to reduce the word count of the article.

Title

The title should reflect the 5C model used to make it catch the attention of the readers.

Abstract

As the study is a quantitative one, the abstract should contain the relevant coefficients showing the relationships between variables.

Introduction

A stronger justification should be given for using university students a representation of young persons, as they are only a fraction of all young persons.

Also, more information should be given about the 5C model. A brief summary of the theory should be added to the paper for readers not already familiar with the model.

Methodology

The methodology is adequate to answer the research objectives.

Discussion and Conclusion

This section is also adequate.

Reviewer #4: Reviewer Comments

1. In the abstract part line 25 the countries and the model connected by and please write separately.

In introduction

1. In your introduction part line 53, which vaccine is different? Please specify

2. The paragraph stated from line 74- 80 seems like discussion and it is better to take it to discussion part of your study.

Materials and methods

1. In material and methods part in line 135-141 you say that the data were collected on two survey and you collect data first from 10 countries. So what are those 10 countries and do the three countries namely (Netherland, Belgium and Portugal) include in the first survey or not? Please specify.

2. In line 151 you stated that your total sample size is 1,137 which is obtained from three countries (Netherlands N= 195, Belgium N= 745, and Portugal N=294) the total value here is 1,234 which is different from the sample size you stated before. How? Specify you sample size clearly.

3. The sampling method you used is not clear. Please write what type of sampling method you use.

4. In line 206 you write r=0.62 and p<0.01. What are r and p? If r is correlation coefficient how could you do correlation likert skale data?

Result

1. On your result part line 329-341 the descriptive statistics about COVID-19 vaccination intention of students is not similar with the result of the bar graph, in the bar graph in fig 3 the highest frequency is “defiantly not” but you interpretation is different.

2. In line 557 You use the model OLS regression for intention of students for COVID-19 vaccination on independent variables. But you did not mention model adequacy please write how could you check your model is adequate.

3.

6. PLOS authors have the option to publish the peer review history of their article (what does this mean?). If published, this will include your full peer review and any attached files.

Reviewer #1: No

Reviewer #2: No

Reviewer #3: No

Reviewer #4: No

---

## [Author Response · Author response to Decision Letter 0]

12 Jul 2021

Dear Dr. Camelia Delcea,

Thank you for allowing us to revise and resubmit our manuscript. Below we list the comments from the reviewers in bold text, followed by our responses in plain text.

We believe that the changes made have significantly improved the manuscript. In our response to each comment, we outline how we have addressed the issues raised by the reviewers. We hope that you and the reviewers agree that the revised version of the manuscript addresses the concerns raised in a transparent and satisfactory way.

Best regards,

The authors

Reviewer #1:

1.1 the authors should be write the abbreviation lists example: COVID-19,UK,IBM,SPSS etc.

Thank you for pointing this out. We have scanned the document and removed unnecessary use of abbreviations. Moreover, we now introduce COVID-19 as abbreviation of coronavirus disease 2019 at the start of the paper. We believe that the use of other abbreviations in the current version is so limited that a list is not necessary. We hope you agree.

1.2 the authors give a gap or the space appears the line of the text in the references.

We no longer indent the first line of the references. Thank you. 

1.3 clarify the statistical model used for this study.

We have extended the description of the statistical methods used. We hope this makes our analyses easier to follow. 

1.4 how to determine the sample size and which sampling technique is applied?

We have added more information on the sampling process to the paper. As you know, we use data of two surveys (T1, April-May 2020, and T1, December 2020). For our study, we make use of data collected from participants that responded both at T1 and T2. In the paper, we now describe the sampling process at T1 per country, sample sizes that were acquired in our first survey (T1), the number of students sharing their e-mail address for potential participation in a follow-up and finally the response rate acquired at T2 which led to the final sample used in our study.

Conducting a power analysis using G*power 3.1.9.7 to estimate the required sample size given an effect size of .02, alpha of .05, a power of .90, and 11 predictors (our most extensive model in the paper), we find that a sample of 1,053 participants would be required. Our sample consists of 1,137 participants, which meets this requirement. We admit that we did not conduct this power analysis upfront. However, we made a motivated decision in only inviting Dutch, Belgian and Portuguese participants for follow-up, as these countries were the only countries for with a sufficient number of students indicated to be willing to participate again at T1.

We would like to express our gratitude to you for reviewing our manuscript and helping to identify weaknesses in the earlier version. We hope that the revised version of the manuscript addresses the concerns you have raised in a transparent and satisfactory way. 

/The authors 

Reviewer #2: 

The authors describe partial results from the Erasmus University Rotterdam International COVID-19 Student Survey (EURICSS), related specifically to vaccination intention in university students from three countries: the Netherlands, Belgium and Portugal. They use the questions related to scales measuring the 5C model and some personality trait questions, along with a measure of vaccination intention, to perform a regression analysis and a mediation analysis.

The paper is written in a clear manner and presents enough details about the definition of each of the variables, and how they were measured in the survey. 

Two things are of concern in my opinion: 

2.1 The authors are not using a probabilistic sample, or at least it was not explained as such in the methods section, and thus using or calculating standard errors and p-values for the regression model and the mediation analysis is not entirely justifiable.

Thank you for this remark. As this point is also addressed in several comments below, we choose to extensively address it in response to the present comment. 

We agree that the use of a non-probabilistic sample is a limitation of our study and that, strictly speaking, the use of inferential statistics is not completely justifiable (Copas & Li, 1997; Smith, 1983). We now more prominently acknowledge this limitation in our paper and have moderated our language and claims. Firstly, we do not longer generalize the results of our sample to all young adults. Moreover, we are less strong in our claims on the generalizability of the results to all students. In social science research such as the present study - involving participants who we have to recruit and ask for permission -, we almost never use pure probabilistic samples (they are more or less probabilistic) and we always have to deal with some form of sampling bias. Completely disregarding our study because of the use inferential statistics would not be in line with the tradition in social sciences. Several studies have shown that samples gathered using Internet methods are at least as diverse as many of the sample used in psychological research and are usually not maladjusted (e.g. Gosling et al., 2004). We believe that there are multiple reasons why the analyses shown in our paper are still valuable and that bias in our sample is limited. We list them below. 

First, as the first wave of data collection took place during the early phase of the COVID-19 pandemic, our main focus was to collect timely data capturing valuable information related to the COVID-19 pandemic. There was no time to acquire sufficient funds to incentivize students and, thereby, to collect a survey that was not based on voluntary participation. While our sample is not entirely probabilistic, we did use a sampling method which makes our sample close to random and which is more solid than most methods employed in a large share of social sciences research, such as sharing via social media, word of mouth and ‘snowball sampling’ (Christner et al. 2021, Coroiu et al., 2020, Wang et al., 2020 – all published in PLOS One). In our study, during the first wave of data collection, large and representative groups of students were systematically approached in all three countries (Netherlands, Belgium, and Portugal). In contrast to the social media and snowball sampling techniques, this means that we did approach a representative group of students. However, as participation was mostly voluntary, we agree that we must address the limitation of using a non-probabilistic sample in our paper and justify why our sample is valid. In the revised version of our manuscript, we now give an extensive explanation of the sampling method and inform the reader about the limitations of our sample in the methods and discussion section. See below the text that was added to the manuscript related to this (part 3 and part 4).

Second, because participation in the survey was on a voluntary basis, we believe that the only bias that may arise in our sample is that we recruited students with high levels of social desirability. In other words, it is possible that students who are more agreeable and showing socially desirable behaviour were more likely to participate and also remain in the study during follow-up than students that did not participate. To control for this aspect, we collected data on the level of social desirability of students using the 13-item short form (C) of the Marlow-Crowne Social Desirability Scale (SDS) during the T2 survey (Crowne & Marlowe, 1960; Reynolds, 1982). SDS’s have been advocated as a means to check the robustness of results based on self-report data (Van de Mortel, 2008). Therefore, we repeated all analyses presented in the paper (Tables 2-7) controlling for social desirability by including scores on the short form SDS and find that all our conclusions presented in the paper stay the same. For the reviewer’s information, the results are presented in Tables 1-6 below under the header ‘Robustness Check - Social Desirability Scale’ (part 1).

Third, while the response rate of our follow-up study was very high (39.2%) and therefore it is less likely to be affected by non-response bias, we conducted additional analyses to compare some demographic and general variables between the T1 and T2 sample. To test whether the participants who participated in the follow-up (of which we use the data in our study) differ from those who only participated at T1, we compared the T1 sample of Belgian, Portuguese, and Dutch students excluding those that participated during follow-up (T2) and compared them to this T2 sample. Continuous variables were compared using independent samples t-tests (Table 7), whereas categorical variables were compared using Pearson Chi-square tests (Table 8). The results of these analyses are presented in part 2 (‘Robustness check – follow-up respondents versus T1 only respondents’).

Overall, we see that T1 and T2 samples did not differ regarding age, government trust, and risk perception of COVID-19 for family and friends. With respect to gender, although the follow-up sample contains less females, the difference is not statistically significant at 5% level. As females were overrepresented in the first wave, the relative decrease in female participants makes the sample we use more representative with respect to gender. Finally, we only find a small difference with respect to risk perception of COVID-19 for oneself. The follow-up sample had a lower risk perception of COVID-19 for oneself than the group that did not participate in T2 (p=.03). We conclude that the differences between the two samples are limited to lower risk perception of COVID-19 for oneself.

In conclusion, we believe that our results are valuable and can be published in their current form. As mentioned, we do agree that we should have been clearer about our sampling method and the accompanying limitations. We now address this both in the methods and discussion (see below for the text that was added to the manuscript (part 3 and 4)). Finally, as indicated above, we have moderated our language throughout the paper and we no longer draw strong conclusions on the generalizability or our results.

1. Robustness Check - Social Desirability Scale 

In Table 1 below, we present the results of the regression analysis of Table 1 in the paper, excluding and including the SDS. One can see that the effect sizes and significance levels of the variables of interest (5C model) and therefore the conclusions drawn in the paper remain the same when including the SDS.

In Tables 2-6 we present the indirect effects of all mediation analyses both including and excluding social desirability in the analyses. Table 1 corresponds to Table 2 in the paper, Table 2 to Table 3 and so on. For reasons of brevity, we have omitted the 30 regression models underlying these indirect effects (three regression models for each set of indirect effects, as presented in the paper). As one can see from the Tables, the conclusions on the indirect effects of the psychological variables on vaccination intention through the 5C’s stay the same for all 5C’s when including the SDS as additional control variable. The 95% Bias corrected confidence intervals all lead to the same conclusions. As in the paper, we have bold-printed those confidence intervals excluding zero. Moreover, the effect sizes of the indirect effects are virtually the same for all indirect effects. Minor differences with a maximum of .02 are present, only increasing the effect sizes and consequently strengthening the conclusions drawn in the paper.

Table 1. OLS regression analyses with vaccination intention as dependent variable – Including and excluding social desirability

- See Tables in response letter file

Table 2. Mediation analyses with Confidence as mediator, excluding and including social desirability

- See Tables in response letter file

Table 3. Mediation analyses with Calculation as mediator, excluding and including social desirability

- See Tables in response letter file

Table 4. Mediation analyses with Complacency as mediator, excluding and including social desirability

- See Tables in response letter file

Table 5. Mediation analyses with Constraints as mediator, excluding and including social desirability

- See Tables in response letter file

Table 6. Mediation analyses with Collective Responsibility as mediator, excluding and including social desirability

- See Tables in response letter file

2. Robustness Check – follow-up respondents versus T1 only respondents 

Table 7. T-tests comparing follow-up respondents and T1 only respondents

- See Tables in response letter file 

Table 8. Pearson Chi-Square tests comparing follow-up respondents and T1 only respondents 

- See Tables in response letter file

3. Added text Limitation section

“Second, as we did not use a probabilistic sample, the use of inferential techniques is not entirely justifiable [70,71]. While we used a large sample of students from three countries and during the sampling process approached large and representative groups of students, participation was (mostly) on a voluntary basis. Since we expected students with higher levels of social desirability to be more likely to participate, we conducted all analyses controlling for social desirability. The fact that our conclusions remained the same strengthen our belief in the validity of our results. However, it is possible that our sample suffers from other type of non-response bias and that our results should therefore be interpreted with caution.”

4. Added text Methods section 

“As we did not use a completely probabilistic sample, it should be noted that our findings may not be generalizable to all students. However, we believe that as we approached representative and large groups of students, risk of bias mostly arises from voluntary participation. It is therefore probable that students who are more agreeable and show more socially desirable behavior are more likely to join in both surveys. To check whether this has affected our outcomes, we conducted all analyses presented in the paper controlling for scores on the adapted 13-item short (form C) Social Desirability Scale of Marlow-Crowne [25,26]. The use of social desirability scales has been advocated to check the robustness of results by controlling for response bias [27]. Based on these additional analyses, we find that all conclusions drawn in the current study remain the same.”

2.2 All the figures in the paper need to have a better quality: figures 1 and 3 do not have an appropriate resolution to be readable. Fig 3 does not include a clear label on the x axis. Figs 2 and 4 do not include the paths, and thus are not very useful.

Thank you for pointing this out. By accident, the x-axis of Figure 3 was replaced by numbers. This has been corrected. Moreover, something seems to have gone wrong with respect to the paths between uploading and sharing the Figures as they are visible in our images. We have made sure the resolution is higher and we expect the paths to be visible now.

2.3 The description of the results per question should be included in the paper. This table is now in the supporting information, but it could be added to the main article without the correlations, and a heat plot can be used to represent the correlations between variables. This description is specially useful when presenting the regression model results. In this case, Table 1 is giving inferential statistics details (that should be calculated only if this was a probabilistic sample), but it is not very illustrative about 1) the assumptions of the model and 2) the complete description of each of the variables.

We have shifted the table including descriptive statistics of all variables of our study from the supporting information to the manuscript. We agree this information is important in interpreting the regression analyses. We now present mean, standard deviations, range of all variables. Moreover, we decided to include the correlation table instead of the heat plot. Finally, we have added the F-statistic of the regression analysis to the table and now describe the variance inflation factors and R2.

2.4 The mediation analysis are hard to follow without Fig 2 and 4 and/or a model equation. In any case they have the same problem as the regression model: a clear description of the variables (are there extreme values that could be affecting the analysis, for example?) and the fact that the authors are using inferential techniques on what looks like a convenience sample.

We have extended the explanation of the mediation analyses. We hope that the extended explanation together with the Figures (including paths) will clarify the analyses performed. As indicated above, we have now added a description of the variables (range, Mean, SD and correlations) to the paper. 

For a response to your critique on the use of non-probabilistic sample, we refer to our reply to comment 2.1 

2.5 In summary, without a probabilistic sample the inferential results should be taken with a grain of salt. The authors do note in the limitations of the study that the sample might not be completely generalizable to all young adults, but they do interpret their results in a matter that leads to believe that they are generalizable to all the students from these three countries. My question is: if this is the case, and if so, how would the authors argue that this is true.

As indicated above, we no longer generalize our findings to all young adults and are more prudent in our interpretations. We have moderated our language and have addressed the limitations that come with the use our sample in the revised version of our manuscript. For an extensive reply, we refer to our response to comment 2.1 in which we address the use of non-probabilistic sample.

If this is clarified, and the recommendations about the results and figures are implemented, I think this manuscript can be technically sound.

We would like to express our gratitude to you for reviewing our manuscript and helping to identify weaknesses in the earlier version. We hope that the revised version of the manuscript addresses the concerns you have raised in a transparent and satisfactory way.

/The authors

References

Christner, N., Essler, S., Hazzam, A., Paulus, M. (2021). Children’s psychological well-being and problem behavior during the COVID-19 pandemic: An online study during the lockdown period in Germany. PLOS One, 16(6), e0253473. 

Coroiu, A., Moran, C., Campbell, T., & Geller, A. C. (2020). Barriers and facilitators of adherence to social distancing recommendations during COVID-19 among a large international sample of adults. PLOS One, 15(10), e0239795.

Copas, J. B., & Li, H. G. (1997). Inference for non‐random samples. Journal of the Royal Statistical Society: Series B (Statistical Methodology), 59(1), 55-95.

Crowne, D. P., & Marlowe, D. (1960). A new scale of social desirability independent of psychopathology. Journal of Consulting Psychology, 24(4), 349.

Gosling, S. D., Vazire, S., Srivastava, S., & John, O. P. (2004). Should we trust web-based studies? A comparative analysis of six preconceptions about internet questionnaires. American Psychologist, 59(2), 93.

Reynolds, W. M. (1982). Development of reliable and valid short forms of the Marlowe‐Crowne Social Desirability Scale. Journal of Clinical Psychology, 38(1), 119-125.

Smith, T. M. F. (1983). On the validity of inferences from non‐random samples. Journal of the Royal Statistical Society: Series A (General), 146(4), 394-403.

Van de Mortel, T. F. (2008). Faking it: social desirability response bias in self-report research. Australian Journal of Advanced Nursing, The, 25(4), 40.

Wang, H., Xia, Q., Xiong, Z., Li, Z., Xiang, W., Yuan, Y., ... & Li, Z. (2020). The psychological distress and coping styles in the early stages of the 2019 coronavirus disease (COVID-19) epidemic in the general mainland Chinese population: A web-based survey. PLOS One, 15(5), e0233410.

 

Reviewer #3: Comments

The paper examines an interesting area, as it examines the psychological characteristics associated with students’ COVID-19 vaccination intention. Having read through the paper, I do not believe it needs any grammar editing.

Having read through the article, I have the following comments and suggestions.

Overall

3.1 The article has too many words in its current state. The authors should try to reduce the word count of the article. 

Thanks for the hint, you are right. In view of the many constructive suggestions of the referees, the length of the manuscript became slightly longer. We will discuss with the editor how to go about. 

Title

3.2 The title should reflect the 5C model used to make it catch the attention of the readers.

Thank you, we agree that adding the 5C model to the title is important to catch readers’ attention and give a better description of our study. We have changed the full title to: ‘Psychological characteristics and the mediating role of the 5C Model in explaining students’ COVID-19 vaccination intention’. The short title – which should consist of less than 100 characters - is: Psychological characteristics, the 5C Model and students’ COVID-19 vaccination intention’.

Abstract

3.3 As the study is a quantitative one, the abstract should contain the relevant coefficients showing the relationships between variables.

We added the standardized coefficients relating to the relationships between the 5C model and vaccination intention presented in Table 1 to the abstract. However, adding the 95% Bias Corrected-Confidence interval for the other seven relationships presented would make the abstract overly long and would lower readability. 

Introduction

3.4 A stronger justification should be given for using university students a representation of young persons, as they are only a fraction of all young persons.

Upon receiving the comments of the reviewers, we have realized that it is not justifiable to generalize the findings of our study to all young adults. Therefore, we decided to no longer talk about young adults in the paper, but rather discuss our results with respect to the student population. Additionally, we also mention the use of our non-probabilistic sample as a limitation both in the data and discussion section. Finally, we have moderated our language throughout the paper and we no longer draw strong conclusions on the generalizability or our results.

3.5 Also, more information should be given about the 5C model. A brief summary of the theory should be added to the paper for readers not already familiar with the model.

We have extended the explanation of the 5C model in the introduction. We now also explain its aim and more extensively describe the 5C’s. Thank you for this suggestion.

Methodology

3.6 The methodology is adequate to answer the research objectives.

Thank you.

Discussion and Conclusion

3.7 This section is also adequate.

Thank you.

We would like to express our gratitude to you for reviewing our manuscript and helping to identify weaknesses in the earlier version. We hope that the revised version of the manuscript addresses the concerns you have raised in a transparent and satisfactory way.

/The authors

Reviewer #4: Reviewer Comments

4.1 In the abstract part line 25 the countries and the model connected by and please write separately.

Thank you for pointing this out. We have split this sentence into two. 

In introduction

4.2 In your introduction part line 53, which vaccine is different? Please specify

This has been clarified. Thank you.

4.3 The paragraph stated from line 74- 80 seems like discussion and it is better to take it to discussion part of your study.

These lines present previous findings of (the limited range of) studies on COVID-19 vaccination intention. We feel that this information should be briefly addressed in the introduction of our paper. We have tried to make it clearer that we talk about previous literature and not about our own results by adding the bold printed words:

 ‘Regarding COVID-19 vaccination, previous studies have shown that women, younger adults, unemployed individuals and those with a lower socioeconomic status are less likely to get vaccinated [11,19,20]. Moreover, it was recently shown that psychological profiles play a role: vaccine-hesitant and vaccine-resistant individuals are less altruistic, conscientious, more disagreeable, emotionally unstable, and self-interested than are vaccine-acceptant individuals [11]. Finally, higher COVID-19 vaccination intention is associated with more positive general and COVID-19 vaccination beliefs, as well as higher perceived vaccine efficacy and safety [20–22].’ 

Materials and methods

4.4 In material and methods part in line 135-141 you say that the data were collected on two survey and you collect data first from 10 countries. So what are those 10 countries and do the three countries namely (Netherland, Belgium and Portugal) include in the first survey or not? Please specify.

We understand that this should be clarified. We have now added some additional explanation to make sure it is clear that the follow-up survey was conducted with only students from the Netherlands, Belgium and Portugal that also participated in the first wave of data collection. Moreover, we now mention the T1 sample size of these countries, the number of Dutch, Belgian and Portuguese students sharing their e-mail address at T1 and the T2 response rate. We hope sharing this information makes the sampling process clearer. We do not want to specifically mention the other seven countries (which are Spain, Ireland, Italy, Sweden, France, India, Colombia), as we think this would be confusing.

4.5 In line 151 you stated that your total sample size is 1,137 which is obtained from three countries (Netherlands N= 195, Belgium N= 745, and Portugal N=294) the total value here is 1,234 which is different from the sample size you stated before. How? Specify you sample size clearly.

The total sample size is indeed 1,137. We made a mistake in communicating the sample sizes for the countries and this has been corrected. Thanks a lot for noting and pointing this out. Moreover, due to missing values, the sample sizes of analyses can be lower. We now mention this in the method section.

4.6 The sampling method you used is not clear. Please write what type of sampling method you use.

Thank you for this comment. We substantially extended the description of our sampling method.

4.7 In line 206 you write r=0.62 and p<0.01. What are r and p? If r is correlation coefficient how could you do correlation likert skale data?

We indeed present Pearson’s correlation coefficient to justify taking the average of the two items, instead of studying them individually. It is advised to report inter-item correlation when computing composite measures of two items, as coefficient alpha is meaningless in this case (Sainfort & Booske, 2000; Verhoef, 2003). Instead of reporting Pearson’s correlation, we now report Spearman’s rho in a clearer manner. 

Result

4.8 On your result part line 329-341 the descriptive statistics about COVID-19 vaccination intention of students is not similar with the result of the bar graph, in the bar graph in fig 3 the highest frequency is “defiantly not” but you interpretation is different.

An issue occurred with the x-axis of Figure 3. In the previous version the x-axis was by accident replaced by numbers instead of the vaccination intention categories. We have changed this and now Figure 3 corresponds to the text in the descriptive statistics. Our apologies, this was a sloppy mistake. Thank you for noticing.

4.9 In line 557 You use the model OLS regression for intention of students for COVID-19 vaccination on independent variables. But you did not mention model adequacy please write how could you check your model is adequate.

We have now added the F-statistic of our regression model to Table 2, which shows our model is a significant improvement compared to a model without predictors. Moreover, the high R2 value (0.54) shows the 5C model explains a large share of the variation in vaccination intention. Finally, we have described the Variance inflation factors of the model, which all lie between 1.1 and 2.1 indicating that there is no multicollinearity.

We would like to express our gratitude to you for reviewing our manuscript and helping to identify weaknesses in the earlier version. We hope that the revised version of the manuscript addresses the concerns you have raised in a transparent and satisfactory way.

/The authors

References

Sainfort, F., & Booske, B. C. (2000). Measuring post-decision satisfaction. Medical Decision Making, 20(1), 51-61.

Verhoef, P. C. (2003). Understanding the effect of customer relationship management efforts on customer retention and customer share development. Journal of Marketing, 67(4), 30-45.

---

## [Editor Report · Decision Letter 1]

15 Jul 2021

Psychological characteristics and the mediating role of the 5C Model in explaining students’ COVID-19 vaccination intention

PONE-D-21-15022R1

Dear Dr. Wismans,

We’re pleased to inform you that your manuscript has been judged scientifically suitable for publication and will be formally accepted for publication once it meets all outstanding technical requirements.

Kind regards,

Camelia Delcea

Academic Editor

PLOS ONE
---

## [Editor Report · Acceptance letter]

19 Jul 2021

PONE-D-21-15022R1 

Psychological characteristics and the mediating role of the 5C Model in explaining students’ COVID-19 vaccination intention 

Dear Dr. Wismans:

I'm pleased to inform you that your manuscript has been deemed suitable for publication in PLOS ONE. Congratulations! Your manuscript is now with our production department. 

Kind regards, 

on behalf of

Dr. Camelia Delcea 

Academic Editor

PLOS ONE